# Open-world Semi-supervised Generalized Relation Discovery Aligned in a Real-world Setting

**William Hogan**     **Jiacheng Li**     **Jingbo Shang**[*]
Department of Computer Science & Engineering
University of California, San Diego
{whogan,j9li,jshang}@ucsd.edu

## Abstract

Open-world Relation Extraction (OpenRE) has recently garnered significant attention. However, existing approaches tend to oversimplify the problem by assuming that all instances of unlabeled data belong to novel classes, thereby limiting the practicality of these methods. We argue that the OpenRE setting should be more aligned with the characteristics of real-world data. Specifically, we propose two key improvements: (a) unlabeled data should encompass known and novel classes, including negative instances; and (b) the set of novel classes should represent long-tail relation types. Furthermore, we observe that popular relations can often be implicitly inferred through specific patterns, while long-tail relations tend to be explicitly expressed. Motivated by these insights, we present a method called KNoRD (**K**nown and **No**vel **R**elation **D**iscovery), which effectively classifies explicitly and implicitly expressed relations from known and novel classes within unlabeled data. Experimental evaluations on several Open-world RE benchmarks demonstrate that KNoRD consistently outperforms existing methods, achieving significant gains.

## 1 Introduction

Relation extraction (RE) is a fundamental task in natural language processing (NLP), aiming to extract fact triples in the format ⟨*head entity*, *relation*, *tail entity*⟩ from textual data. Open-world RE (OpenRE) is a related research area that focuses on *discovering* novel relation classes from unlabeled data. Recent advancements in OpenRE have demonstrated impressive results by integrating prompting techniques with advanced clustering methods (Zhao et al., 2021; Li et al., 2022b; Wang et al., 2022a). However, current OpenRE methods face limitations due to assumptions about unlabeled data that do not align with the characteristics of real-world datasets. These assumptions include:

(1) the presumption that unlabeled data solely consists of novel classes or is pre-divided into sets of known and novel instances; (2) the absence of negative instances; (3) the random division of known and novel classes in a dataset; and (4) the availability of the ground-truth number of novel classes in unlabeled data.

In this work, we critically examine these assumptions and align the task of OpenRE within a real-world setting. We dispose of simplifying assumptions in favor of new assumptions that align with characteristics of real-world unlabeled data in hopes of increasing the practicality of these methods. We call our setting *Generalized Relation Discovery* and make the following claims:

(a) **Unlabeled data includes known, novel, and negative instances**: Unlabeled data, by definition, lacks labels; we cannot assume it only consists of novel classes or is pre-divided into sets of known and novel instances. Our challenge is to accurately classify known classes and discover novel classes within unlabeled data. Additionally, many sentences with an entity pair do not express a relationship (e.g., negative instances, or the *no relation* class) (Zhang et al., 2017a). Neglecting negative instances in training leads to models with a positive bias, reducing their effectiveness in identifying relationships in real-world data. Hence, we opt to include negative instances in our setting.

(b) **Novel classes are typically rare and belong to the long-tail distribution**: To define known and novel classes, we base our selection process on the intuition that known classes are more likely to be common, frequently appearing relationships. In contrast, unknown, novel classes are more likely to be rare (i.e., long-tail) relationships. Instead of randomly choosing the set of novel classes, we construct data

---

[*]Corresponding author

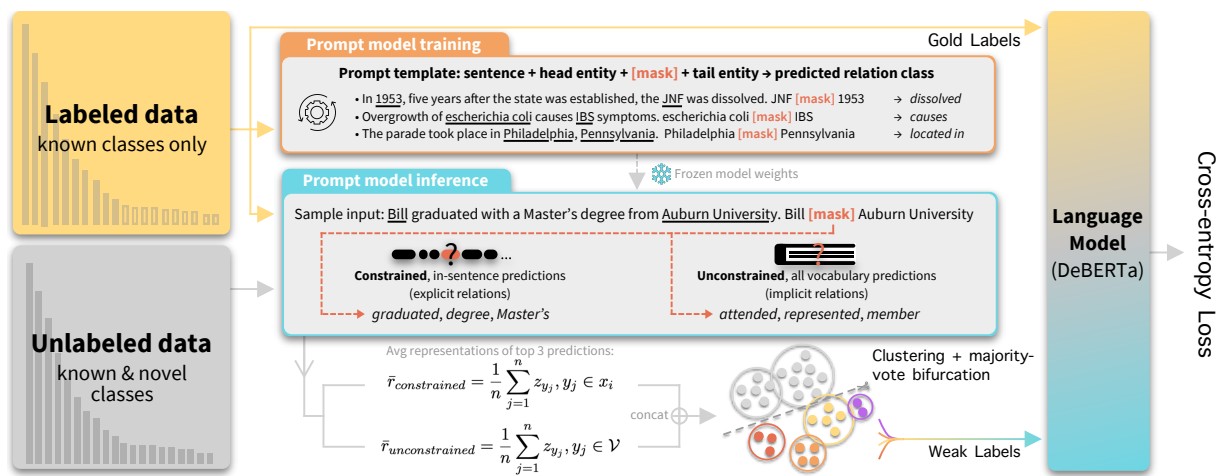

Figure 1: In KNoRD, we use labeled data to train a prompt model to predict relation class names. That model is then used to generate constrained (in-sentence) and unconstrained (all vocabulary) predictions. We average and concatenate representations from the top three constrained and unconstrained predictions. Representations are clustered using Gaussian Mixture Models (GMM) and bifurcated into sets of known and novel instances via a majority-vote. Novel-identified clusters provide weak labels in a cross-entropy training objective.

splits based on class frequency. Although it is possible for frequently appearing classes to be unknown, we deliberately select rare classes for our novel classes to create a more challenging setting. Lastly, without labels, it is impossible to know a priori the ground truth number of novel classes contained within unlabeled data; we do not assume we can access this information in our setting.

Our experimental results show that our proposed setting makes for a difficult task, ripe for advancements and future work.

State-of-the-art approaches in relation discovery leverage a prompt-learning method to predict relation class names which are then embedded into a latent space and grouped via clustering (Zhao et al., 2021; Li et al., 2022b; Wang et al., 2022a). In addition to making simplifying assumptions about unlabeled data, past works use unconstrained predictions for relation class names—that is, the prompt model can predict any word in its vast vocabulary as a relation class name. We observe that relationships in text can be expressed either *explicitly* or *implicitly*. Explicit instances contain class-indicative words, and implicit relationships are inferred via lexical patterns (see Appendix A.1 for examples). In this work, we illustrate the effectiveness of designing a prompt method that optimizes for explicit and implicit relationships by predicting relation class names in two settings: (1) constrained to words found within an

inputted instance; and, (2) unconstrained, where the model can predict any word within its vocabulary. Constrained predictions optimize for explicitly expressed relationships while unconstrained predictions optimize for implicitly expressed relationships. This prompt method forms the backbone of our proposed method, **K**nown and **No**vel **R**elation **D**iscovery (KNoRD), which can effectively classify explicitly and implicitly expressed relationships of known and novel classes from unlabeled data.

Another key aspect of our method is that it clusters labeled and unlabeled data within the same feature-space. Each labeled instance serves as a "vote" for a cluster belonging to the set of known classes. We effectively bifurcate clusters into sets of known and novel classes by employing a majority-vote strategy. Novel-identified clusters are then utilized as weak labels, in combination with gold labels, to train a model via cross-entropy (see Figure 1). This methodology presents an innovative approach to relation discovery in open-world scenarios, offering potential applications across various NLP domains.

The main contributions of this work are:

- We critically examine the assumptions made in OpenRE and carefully craft a new setting, *Generalized Relation Discovery*, that aligns with characteristics of real-world data.
- We propose an innovative method to classify known and novel classes from unlabeled data.

- We illustrate the effectiveness of modeling implicit and explicit relations via prompting.
- We openly provide all code, experimental settings, and datasets used to substantiate the claims made in this paper.[1]

## 2 Related Work

Traditionally, RE methods have focused on a closed-world setting where the extracted relations are predefined during training (Califf and Mooney, 1997; Mintz et al., 2009; Zhang and Wang, 2015; Peng et al., 2017; Qin et al., 2021). However, datasets used for training are rarely complete, and traditional RE cannot capture new relation classes as additional data becomes available. Researchers have proposed various approaches to discover emergent classes to address this issue.

**Open-world RE**: Open-world RE seeks to discover new relation classes from unlabeled data. Instances of relations are typically embedded into a latent space and then clustered via K-Means. These works often simplify the task by assuming all instances of unlabeled data belong to the set of novel classes and that unlabeled data contains no negative instances (Elsahar et al., 2017; Hu et al., 2020; Zhang et al., 2021). More recently, some OpenRE methods have been proposed to predict known and novel classes from unlabeled data (Zhao et al., 2021; Wang et al., 2022a; Li et al., 2022b). However, these works assume unlabeled data comes pre-divided into sets of known and novel instances, that negative instances are removed, and that the number of novel classes is known.

**Open-world semi-supervised learning**: The setting we propose for relation discovery is inspired by open-world semi-supervised learning (Open SSL) proposed by Cao et al. (2022) where the authors use a margin loss objective with adaptive uncertainty to predict known and novel classes from a set of unlabeled images. Besides the difference in domains, their setting differs from ours in that they assume unlabeled data has an equal number of known and novel instances—an assumption we cannot make when working with relation extraction datasets which often have imbalanced, long-tail class distributions (Zhang et al., 2017b; Stoica et al., 2021; Yao et al., 2019; Amin et al., 2022). Furthermore, Cao et al. (2022) assumes unlabeled data only contains positive instances, which does

[1] https://github.com/wphogan/knord

Characteristics of unlabeled data ($\mathcal{D}_u$):

| Setting | Known & Novel Cls | W/Neg. Instances | # Novel is Unk | Novel classes are long-tail | Limiting Assumptions |
|---|---|---|---|---|---|
| Traditional RE | × | × | × | × | Only predicts classes seen in training |
| ZeroShot RE | × | × | × | × | $\mathcal{D}_u$ only contains novel classes |
| Gen. ZeroShot RE | ✓ | × | × | × | Novel classes require definitions |
| Continual RE | ✓ | × | × | × | Novel classes require human annotation |
| Open RE | ✓ | × | × | × | $\mathcal{D}_u$ is pre-divided into known/novel instances |
| Open SSL | ✓ | × | ✓ | × | No negative instances; rand. novel classes |
| **Gen. Rel. Discovery** | ✓ | ✓ | ✓ | ✓ | — |

Table 1: A comparison between our proposed setting, Generalized Relation Discovery, and the existing settings. We remove simplifying assumptions about unlabeled data ($\mathcal{D}_u$) to align the task to the characteristics of real-world data.

not transfer to our task where a prevalence of sentences do not express a relationship (Zhang et al., 2017b).

The setting proposed in Li et al. (2022c) is similar to ours however, negative instances are removed and their known/novel class splits are done randomly instead of by class frequency. Furthermore, their method relies on active learning where human annotators annotate instances of novel classes. The annotations are then used to train a classifier. In contrast, our model and the baselines we evaluate do not require human annotation for novel classes.

Table 1 qualitatively compares our proposed setting, Generalized Relation Discovery, to related settings. Additional related works are discussed in Appendix A.2.

## 3 Problem Statement

The task of Generalized Relation Discovery is to simultaneously classify instances of known relation classes and discover novel relation classes from unlabeled data, given a labeled dataset with a set of known classes.

We construct a transductive learning setting where we assume both labeled data $\mathcal{D}_l = \{x_i, y_i\}, y_i \in \mathcal{C}^K$ and unlabeled data $\mathcal{D}_u = \{x_i\}$ are given as inputs. All classes in $\mathcal{D}_l$ are considered known classes $\mathcal{C}^K = \{c_k^1, ..., c_k^i\}$ where $i = |\mathcal{C}^K|$ is the number of known relation classes. $\mathcal{D}_u$ consists of instances from $\mathcal{C}^K$ as well as instances of novel classes $\mathcal{C}^N = \{c_n^1, ..., c_n^j\}$ where $j = |\mathcal{C}^N|$ is the

number of novel relation classes. The known and novel relation class sets are constructed as non-intersecting sets (i.e., $\mathcal{C}^K \cap \mathcal{C}^N = \emptyset$). Following Cao et al. (2022), we assume there is no distribution shift between the labeled and unlabeled data (i.e., known relation classes found in labeled data also occur in the unlabeled data).

We formulate our task in the following way:

$$\mathcal{Y} = \left\{ y_j^{c_i}, y_j'^{c_i'} \mid x_j \right\}, c_i \in \mathcal{C}^K, c_i' \in \mathcal{C}^N, x_j \in \mathcal{D}_u$$

where $x_j$ are unlabeled instances (e.g, sentences), $y$ and $y'$ denote known and novel predictions, $c$ and $c'$ denote known and novel classes, respectively, and $\mathcal{Y}$ is the set of all predictions for instances in $\mathcal{D}_u$. Each sentence $x_j$ contains an entity pair—a head entity $e_1$ and a tail entity $e_2$—and the predicted relationship $y_j$ links the two entities, producing a fact triplet $\langle e_1, y_j, e_2 \rangle$.

## 4 Method

KNoRD consists of four discrete stages: (1) prompt-based training, (2) constructing semantically-aligned relation representations, (3) clustering with majority-vote bifurcation, and (4) classification. We describe each stage in detail in the following subsections.

### 4.1 Prompt-based Training

We leverage the instances of labeled data to train a language model to predict the linking relationship between an entity pair found in a sentence via prompting. Specifically, given a sentence $x_j$, we construct a prompt template $\mathcal{T}(\cdot) = \langle e_1 \rangle$ [MASK] $\langle e_2 \rangle$ where $e_1$ and $e_2$ are two entities in $x_j$.[2] The template is appended to $x_j$ to obtain contextualized relation instance. Then, a masked language model (e.g., BERT (Devlin et al., 2019)) learns to predict the masked tokens between two entities. To alleviate the model overfitting on the relation token vocabulary, we randomly mask 15% of tokens in $x_j$, and the model is jointly trained to predict the masked tokens in sentences and masked relation names.

During inference, we feed the contextualized relation instance with only masked relations[3] to the model and predict the masked token. Top-ranked tokens predicted for [MASK] from the model are

---

[2] The number of [MASK] tokens is the same as the number of tokens in the relation name.

[3] We use only one [MASK] between entities for inference.

used to create semantically-aligned representations in the subsequent stage.

### 4.2 Semantically-aligned Representations

Leveraging our observation that relationships are expressed explicitly or implicitly, we construct two settings for our prompt model: constrained and unconstrained predictions. Constrained predictions are [MASK] predictions ($y_i$) constrained to words found within the inputted instance $x_i$, i.e., $y_i \in x_i$ where $x_i$ is the input sentence with words $\{w_1, \ldots, w_n\}$. In this setting, top tokens in the inputted instance are used to optimize for *explicitly expressed* relationships. In the unconstrained setting, we allow the model to use any word in its entire vocabulary ($\mathcal{V}$) to predict the name of the relationship, i.e., $y_i \in \mathcal{V}$, optimizing for *implicitly expressed* relationships.

We use the hidden representations of the top three tokens in each setting to construct the following representations:

$$\bar{r}_{i\,constrained} = \frac{1}{n} \sum_{j=1}^{n} z_{y_j}, y_j \in x_i \qquad (1)$$

$$\bar{r}_{i\,unconstrained} = \frac{1}{n} \sum_{j=1}^{n} z_{y_j}, y_j \in \mathcal{V} \qquad (2)$$

where $n = 3$ and $z_{y_j}$ is the $j^{th}$ embedded representation ($z_{y_j} \in \mathbb{R}^D$) of the prediction corresponding to instance $x_i$ from a phrase embedding model (Li et al., 2022a). Note, we do not use the prompting model to produce $z$ because this model is trained on only known classes and tends to overfit on known classes even if random tokens are masked and predicted in $x_i$ during training.

Our final relationship representation is constructed by combining the constrained and unconstrained representations:

$$r_i = \langle \bar{r}_{i\,constrained}, \bar{r}_{i\,unconstrained} \rangle \qquad (3)$$

where $\langle , \rangle$ represents concatenation. The combined representation $r_i$ models explicitly and implicitly expressed relationships in sentence $x_i$.

### 4.3 Clustering with Majority-vote Bifurcation

Relationship representations from Equation 3 are clustered via Gaussian Mixture Models (GMM). To improve the quality of clusters, we adjust the cluster member according to their entity meta-type

pairs (e.g. [*human*, *organization*], etc.). Specifically, we select the top 30% of relation instances in each cluster and use the major entity meta-type pair of them as the meta-type of the cluster. Then, all relation instances are adjusted to the nearest cluster with the same meta-type.

We cluster instances from both labeled and unlabeled data into the same feature-space. Intuitively, since all labeled instances are instances of known classes, each labeled instance acts as a "vote" voting for a cluster that corresponds to a known class. We tally the votes of all the labeled instances and use the results to bifurcate the set of clusters into two subsets of known-class clusters $G^K$ and novel class clusters $G^N$ such that $G^K \cap G^N = \emptyset$. We call this method "majority-vote bifurcation" and use the novel-identified clusters $G^N$ as weak labels for the subsequent classification module.

### 4.4 Relation Classification

In the final stage of KNoRD, we use gold labels from labeled data and weak labels generated from the method described in Section 4.3 to train a relation classification model.

Since the clusters generate weak labels of varying degrees of accuracy, we select the top $P\%$ of weak labels for each cluster in $G^N$. In our model, we set $P = 15$. We retrospectively explore the effects of different $P$ values and report the performance in Appendix A.5. We observe that the optimal value for $P$ varies across datasets. We leave developing an advanced method of determining $P$ for future work.

For each relationship instance, we follow Soares et al. (2019) and wrap head and tail entities with span-delimiting tokens. We construct entity and relationship representations following Hogan et al. (2022). For sentence $x_i$, we use a pre-trained language model, namely DeBERTa (He et al., 2021),[4] as an embedding function to obtain feature representations for relationships.

We encode $x_i$ and obtain the hidden states $\{\mathbf{h}_1, \mathbf{h}_2, \dots, \mathbf{h}_{|x_i|}\}$. Then, *mean pooling* is applied to the consecutive entity tokens to obtain representations for the head and tail entities ($e_1$ and $e_2$, respectively). Assuming $n_{\text{start}}$ and $n_{\text{end}}$ are the start and end indices of entity $e_1$, the entity repre-

---

[4] https://huggingface.co/microsoft/deberta-base

sentation is:

$$\mathbf{m}_{e1} = \text{MeanPool}(\mathbf{h}_{n_{\text{start}}}, \dots, \mathbf{h}_{n_{\text{end}}}) \quad (4)$$

To form a relation representation, we concatenate the representations of two entities $e_1$ and $e_2$: $\mathbf{r}_{e1e2} = \langle \mathbf{m}_{e1}, \mathbf{m}_{e2} \rangle$. The relation representations are sent through a fully-connected linear layer which is trained using cross-entropy loss:

$$\mathcal{L}_{\text{CE}} = -\sum_{i=1}^{N} y_{o,i} \cdot \log\left(p\left(y_{o,i}\right)\right) \quad (5)$$

where $y$ is a binary indicator that is 1 if and only if $i$ is the correct classification for observation $o$, $p(y_{o,i})$ is the Softmax probability that observation $o$ is of class $i$, and $N$ is the number of classes. Predictions from Equation 5 are mapped to ground-truth classes using the Hungarian Algorithm (Kuhn, 1955) (see Appendix A.6.2 for more details).

Li et al. (2022b) show that setting the number of novel classes to a large number corresponds to fine-grained novel class predictions which, depending on the task and desired outcome, can be grouped into more general classes via abstraction. Since we do not assume the ground-truth number of novel classes is available, we use a relatively high number of novel classes equal to twice the number of known classes ($2 \times |\mathcal{C}^K|$), where $|\mathcal{C}^K|$ is the number of known classes found in the labeled data. The $N$ in Equation 5 is set to $|\mathcal{C}^K| + (2 \times |\mathcal{C}^K|)$. We leave developing an automated method for class abstraction for future work.

## 5 Datasets

We evaluate KNoRD on three RE datasets: TACRED (Zhang et al., 2017b), ReTACRED (Stoica et al., 2021), and FewRel (Han et al., 2018). For each dataset, we first construct splits of known and novel classes based on class frequency, assigning the top 50% most frequent relation classes to the set of known classes ($C^K$) and the lower 50% to the set of novel classes ($C^N$) (see Figure 2). Since FewRel is a balanced dataset with relationships defined from a subset of Wikidata relationships, we obtain real-world class frequencies based on their frequency within Wikidata. For more details on our FewRel pre-processing steps, see Appendix A.3.

All instances of novel classes are combined with a random sample of 15% of known-class instances

| | FewRel | ReTACRED | TACRED |
|---|---|---|---|
| # Known Classes | 40 | 19 | 20 |
| # Novel Classes | 40 | 20 | 21 |
| # Negative Instances (labeled \| unlabeled) | 0[†] | 57,777 (49,110 \| 8,667) | 84,491 (71,817 \| 12,674) |
| # Labeled Instances* | 23,626 | 25,478 | 15,388 |
| # Unabeled Instances* (known \| novel) | 32,374 (4,374 \| 28,000) | 8,212 (4,474 \| 3,738) | 6,385 (2,865 \| 3,520) |

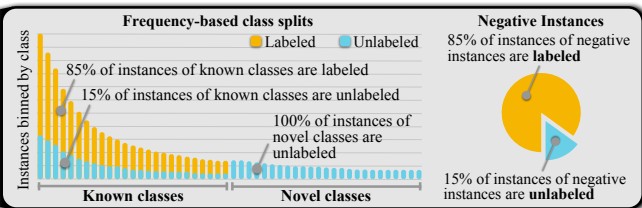

Figure 2: Data splits used in our Generalized Relation Discovery setting. Given $n$ total classes in a dataset, the set of known classes are the top $\lfloor n/2 \rfloor$ most frequent classes. Remaining classes are placed into the set of novel classes. Labeled data consists of 85% of instances from known classes. Unlabeled data contains 15% of instances of known classes and 100% of the instances from novel classes (*numbers do not include negative instances, [†]since FewRel has no annotated negative instances, we augment the dataset using negative instances from ReTACRED[5]).

to form the unlabeled dataset $D_u$. The remaining 85% of known-class instances are used to create the labeled dataset $D_l$.

We include two versions of each dataset: with and without negative instances (e.g., the *no relation* class). The setting with negative instances best mirrors real-world data; however, as our experiments show, discovering novel classes in a sea of negative instances is difficult. We include results from both settings and the setting with negative instances will be ripe for advancement and future work.

Our focus is to evaluate methods on data with no distribution drift (i.e., known classes from training occur in the unlabeled data along with novel classes). We leave an evaluation on out-of-distribution datasets (Gao et al., 2019; Bassignana and Plank, 2022) to future work.

# 6 Experiments

We compare KNoRD to state-of-the-art OpenRE baselines: (1) RoCORE (Zhao et al., 2021), (2) MatchPrompt (Wang et al., 2022a),[6] (3) TABs (Li et al., 2022b). Since OpenRE methods cannot naturally operate within the Generalized Relation Discovery setting, we extend the OpenRE baselines in the following ways:

(i) **RoCORE′**, **MatchPrompt′**, **TABs′**: Given that OpenRE methods cannot identify previously seen (known) classes mixed with novel classes in unlabeled data, we evaluate their performance on novel classes and propose a method to adapt them for seen classes. To achieve this, we treat all classes as novel classes, enabling these methods to effectively cluster the unlabeled data. Subsequently,

we employ the Hungarian Algorithm (Kuhn, 1955) to match some discovered classes to known classes from labeled data, facilitating performance evaluation on the known classes.

(ii) **RoCORE[†]**, **MatchPrompt[†]**, **TABs[†]**: Many leading OpenRE models assume unlabeled data comes pre-divided into sets of known and novel instances (Zhao et al., 2021; Li et al., 2022b; Wang et al., 2022a). A natural extension of these methods is to prepend a module that segments unlabeled data into known and novel instances. We pre-train models on known classes and then generate confidence scores for each unlabeled instance. We use the softmax function as a proxy for confidence (Hendrycks and Gimpel, 2016) and set the confidence threshold equal to the mean confidence from labeled instances of known classes. Instances with confidence scores below the threshold are assigned to novel classes. We report the accuracy of this method in bifurcating unlabeled data in Appendix A.4.4.

**ORCA**: We include OCRA (Cao et al., 2022), a computer vision model developed for a similar generalized open-world setting. OCRA is the only model architecture in our experiments that can predict known and novel classes from unlabeled data and, thus, requires no modification, beyond adaptation to the RE task, to function within our proposed setting. For more details about adapting ORCA to predict relationships, see Appendix A.4.5.

**GPT 3.5**: Given the zero-shot learning capabilities of Large Language Models (Kojima et al., 2022), we also include GPT 3.5 (OpenAI, 2021)[7] as a baseline. To assess GPT 3.5, we leverage in-context learning—we provide examples of extracted relationships and a list of known relation classes. We

---

[5]See Appendix A.3 for more details.

[6]At the time of writing, the authors of MatchPrompt have not released the code for their method. The method used in this paper is from our own implementation.

[7]*We use the gpt-3.5-turbo-0301* version of ChatGPT via OpenAI's API.

instruct the model to predict the most appropriate relation class name or suggest a novel class name when an instance does not fit within the set of known classes (see Appendix A.4.1 for details). **GPT 3.5 +*cos***: Since responses from GPT 3.5 may not align perfectly with ground-truth labels, we use DeBERTa to map the responses ($y_i$) to ground-truth class names by embedding the predictions and the ground-truth class names ($z_{y_i}$ and $\mathbf{Z}_{gt} \in \mathbb{R}^{D \times (|C^K| + |C^N|)}$, respectively) and identifying the ground-truth class that exhibits the highest cosine similarity with the predicted class:

$$y_m = argmax \left( \frac{z_{y_i} \cdot \mathbf{Z}_{gt}}{\max \left( \|z_{y_i}\|_2 \cdot \|\mathbf{Z}_{gt}\|_2 , \epsilon \right)} \right)$$

where $y_{m_i}$ is the mapped prediction of prediction $y_i$, and $\epsilon = 1e-8$. We denote the GPT 3.5 baseline with mapped predictions as "GPT 3.5 +*cos*."

For all our baselines, we use identical settings for the number of known and novel classes and, save GPT 3.5, we use the same pre-trained model (He et al., 2021) as a base model and map predictions to ground-truth classes via the Hungarian Algorithm. We use Micro-F1 scores to assess relation classification performance and report overall performance as well as performance on known and novel classes to assess a model's ability to identify each class type from unlabeled data.

## 7 Results

Our proposed method, KNoRD, outperforms the baseline models in all metrics (Table 2). We observe that the ORCA baseline demonstrates strong overall performance and the OpenRE methods (RoCORE′, MatchPrompt′, TABs′) yield diverse results, which we attributes to the differences in underlying architectures. Models such as TABs and MatchPrompt incorporate clustering methods that effectively develop relationship representations in an unsupervised setting. In contrast, RoCORE relies more heavily on supervised training to form high-quality relationship representations. This distinction is evident in our confidence-based adaptations (RoCORE†, MatchPrompt†, TABs†), where pre-dividing the unlabeled data benefits RoCORE significantly while the results for MatchPrompt and TABs are mixed.

We observe that GPT 3.5 underperforms in this setting. Although mapping responses to ground-truth classes (GPT 3.5 +*cos*) yields a slight performance

boost, the model still performs poorly relative to our other baselines. Given the unsatisfactory results from GPT 3.5 in our simplified experiment setting without negative instances, we decide to exclude it from the more challenging setting where negative instances are present. We conclude that more advanced techniques are required to enable GPT 3.5 to accurately classify and discover relationships from textual data. A deeper examination of GPT 3.5's performance is provided in Appendix A.4.3.

In the setting with negative instances, all methods struggle to identify novel relation classes indicating the difficulty of discovering new classes among instances with *no relation*. We attribute the lower overall performance using TACRED compared to ReTACRED to TACRED's wrong labeling problem (Stoica et al., 2021).

The relatively small drop in performance of all models between FewRel with and without negative instances can be attributed to FewRel lacking annotated negative instances, so we artificially augment the data with negatives from ReTACRED. We posit that the models can exploit the slight difference in the distribution of the augmented negative instances, thus reducing the task's difficulty. These results emphasize the importance of future RE dataset creation efforts in annotating negative instances.

### 7.1 Ablations

We conduct ablation studies to better understand the relative importance of each design choice behind KNoRD.

- **KNoRD w/constrained**: We only use constrained predictions from the prompt model to construct relationship representations (Equation 1) and keep all other modules unchanged for this ablation.
- **KNoRD w/unconstrained**: Similar to the aforementioned ablation, but we only leverage unconstrained predictions to construct relationship representations (Equation 2) for the GMM module.
- **KNoRD without CE**: We remove the cross-entropy (CE) module from KNoRD and allow the GMM to predict relation classes directly. We remap cluster predictions to ground-truth classes using the Hungarian Algorithm.

Table 3 shows the performance of all ablation ex-

| | Model | ReTACRED | | | TACRED | | | FewRel | | |
|---|---|---|---|---|---|---|---|---|---|---|
| | | F1 (all) | F1 (known) | F1 (novel) | F1 (all) | F1 (known) | F1 (novel) | F1 (all) | F1 (known) | F1 (novel) |
| | Fully supervised | 0.963 | 0.966 | 0.938 | 0.925 | 0.939 | 0.849 | 0.912 | 0.902 | 0.922 |
| w/o neg. instances | ORCA | 0.622 | 0.870 | 0.325 | 0.521 | 0.719 | 0.360 | 0.411 | 0.398 | 0.414 |
| | RoCORE′ | 0.117 | 0.174 | 0.049 | 0.101 | 0.126 | 0.081 | 0.069 | 0.002 | 0.080 |
| | RoCORE† | 0.578 | 0.846 | 0.257 | 0.380 | 0.629 | 0.177 | 0.352 | 0.314 | 0.358 |
| | MatchPrompt′ | 0.558 | 0.601 | 0.506 | 0.627 | 0.660 | 0.600 | 0.397 | 0.398 | 0.397 |
| | MatchPrompt† | 0.682 | 0.826 | 0.509 | 0.585 | 0.758 | 0.444 | 0.573 | 0.655 | 0.560 |
| | TABs′ | 0.674 | 0.816 | 0.505 | 0.595 | 0.724 | 0.489 | 0.535 | 0.215 | 0.585 |
| | TABs† | 0.312 | 0.304 | 0.320 | 0.298 | 0.294 | 0.302 | 0.541 | 0.370 | 0.568 |
| | GPT 3.5 | 0.279 | 0.450 | 0.075 | 0.277 | 0.482 | 0.111 | 0.098 | 0.313 | 0.064 |
| | GPT 3.5 +*cos* | 0.283 | 0.453 | 0.079 | 0.280 | 0.483 | 0.114 | 0.101 | 0.314 | 0.066 |
| | **KNoRD** | **0.793** | **0.927** | **0.632** | **0.718** | **0.860** | **0.603** | **0.606** | **0.662** | **0.597** |
| | Fully supervised | 0.969 | 0.974 | 0.922 | 0.748 | 0.737 | 0.814 | 0.911 | 0.904 | 0.918 |
| w/neg. instances | ORCA | 0.570 | 0.737 | 0.203 | 0.354 | 0.453 | 0.082 | 0.402 | 0.373 | 0.406 |
| | RoCORE′ | 0.362 | 0.244 | 0.302 | 0.048 | 0.018 | 0.049 | 0.064 | 0.000 | 0.072 |
| | RoCORE† | 0.555 | 0.758 | 0.181 | 0.391 | 0.619 | 0.086 | 0.360 | 0.322 | 0.366 |
| | MatchPrompt′ | 0.416 | 0.429 | 0.210 | 0.274 | 0.253 | 0.148 | 0.529 | 0.287 | 0.542 |
| | MatchPrompt† | 0.532 | 0.536 | 0.372 | 0.330 | 0.339 | 0.173 | 0.563 | 0.622 | 0.552 |
| | TABs′ | 0.550 | 0.615 | 0.329 | 0.358 | 0.334 | 0.233 | 0.511 | 0.253 | 0.551 |
| | TABs† | 0.186 | 0.158 | 0.123 | 0.143 | 0.116 | 0.088 | 0.532 | 0.307 | 0.549 |
| | **KNoRD** | **0.677** | **0.832** | **0.434** | **0.536** | **0.685** | **0.355** | **0.574** | **0.689** | **0.555** |

Table 2: F1-micro scores reported on unlabeled data with and without negative (e.g., *no relation*) instances. *F1 (known)* and *F1 (novel)* report performance on ground-truth known and novel classes, respectively. OpenRE models are extended to operate in the Generalized Relation Discovery setting (see Section 6 for details). All scores average five runs except the GPT 3.5 scores which are resultant from a single run.

| | Method | ReTACRED | | | TACRED | | | FewRel | | |
|---|---|---|---|---|---|---|---|---|---|---|
| | | F1 (all) | F1 (known) | F1 (novel) | F1 (all) | F1 (known) | F1 (novel) | F1 (all) | F1 (known) | F1 (novel) |
| w/o neg. | KNoRD w/constrained | 0.759 | 0.886 | 0.606 | 0.653 | 0.792 | 0.539 | 0.491 | 0.650 | 0.466 |
| | KNoRD w/unconstrained | 0.776 | 0.918 | 0.607 | 0.633 | 0.815 | 0.485 | 0.575 | 0.648 | 0.564 |
| | KNoRD w/o CE | 0.413 | 0.354 | 0.487 | 0.489 | 0.402 | 0.559 | 0.509 | 0.371 | 0.529 |
| | KNoRD | 0.793 | 0.927 | 0.632 | 0.718 | 0.860 | 0.603 | 0.606 | 0.662 | 0.597 |
| w/neg. | KNoRD w/constrained | 0.568 | 0.718 | 0.305 | 0.498 | 0.699 | 0.246 | 0.544 | 0.665 | 0.525 |
| | KNoRD w/unconstrained | 0.507 | 0.700 | 0.119 | 0.490 | 0.686 | 0.241 | 0.507 | 0.700 | 0.119 |
| | KNoRD w/o CE | 0.282 | 0.300 | 0.258 | 0.307 | 0.313 | 0.301 | 0.507 | 0.246 | 0.554 |
| | KNoRD | 0.677 | 0.832 | 0.434 | 0.536 | 0.685 | 0.355 | 0.574 | 0.689 | 0.555 |

Table 3: Ablation experiments varying relationship representation methods used in KNoRD, as well as removing the cross-entropy module and using cluster predictions directly ("w/o CE").

periments. Without CE, KNoRD performs poorly, emphasizing the need for selecting high-quality weak labels to train a CE module. Constrained predictions from the prompt model outperform unconstrained predictions in 4 out of 6 experiments in predicting novel classes, indicating their suitability for rare, long-tail relations. Combining constrained and unconstrained predictions in KNoRD yields the best overall results, demonstrating the effectiveness of optimizing the prompt method to capture explicit and implicit relationships.

We also manually evaluate the accuracy of our prompt method in predicting relation class names. We evaluate the alignment of the top one and top three constrained and unconstrained predictions with ground-truth class names (Figure 3). Constrained predictions, designed to model explicit relationships, are generally more accurate for long-tail relation classes. Conversely, unconstrained predictions perform better on common relationships.

This observed phenomenon roughly aligns with Zipf's Law (Zipf, 1936), indicating that rare concepts and relations are more likely to appear explicitly in a long-form manner. In contrast, common relations tend to be expressed in a compressed form (e.g., implicitly). This insight lends additional evidence to designing a prompt method that captures explicit and implicit relationships.

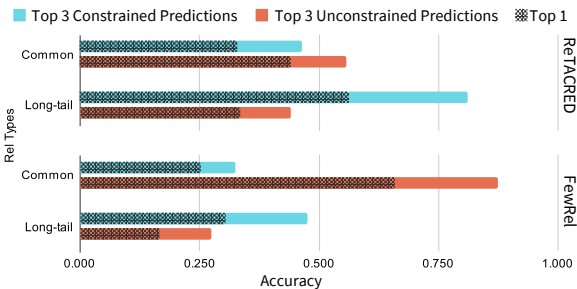

Figure 3: Accuracy of top 1 and top 3 predictions from our prompt method in two settings: constrained (in-sentence words) and unconstrained (all vocabulary) predictions. Unconstrained predictions perform well with common relationships, while constrained predictions perform well on long-tail relationships.

## 8 Conclusion

In this work, we address the limitations of existing approaches in OpenRE and introduce the Generalized Relation Discovery setting to align the task to characteristics of data found in the real-world. By expanding the scope of unlabeled data to include known and novel classes, as well as negative instances, and incorporating long-tail relation types in the set of novel classes, we aim to enhance the practicality of OpenRE methods.

Furthermore, we propose KNoRD, a novel method that effectively classifies explicitly and implicitly expressed relations from known and novel classes within unlabeled data. Through comprehensive experimental evaluations on various Open-world RE benchmarks, we demonstrate that KNoRD consistently outperforms existing methods, yielding significant performance gains. These results highlight the efficacy and potential of our proposed approach in advancing the field of OpenRE and its applicability to real-world scenarios.

## 9 Limitations

The limitations of our method are as follows:

1. Our method requires human-annotated data, which is expensive and time-consuming to create.
2. Our method cannot automatically determine the ground truth number of novel classes in unlabeled data. We leave this to future work.
3. Our method focuses on sentence-level relation classification, and without further testing, we cannot claim these methods work well for document-level relation classification.

4. The low F1 scores of our model and all leading OpenRE models within our experiments with negative instances highlight an area for growth in future works.

## 10 Ethical Concerns

We do not anticipate any major ethical concerns; relation discovery is a fundamental problem in natural language processing. A minor consideration is the potential for introducing certain hidden biases into our results (i.e., performance regressions for some subset of the data despite overall performance gains). However, we did not observe any such issues in our experiments, and indeed these considerations seem low-risk for the specific datasets studied here because they are all published.

## Acknowledgements

This work is sponsored in part by NSF CAREER Award 2239440, NSF Proto-OKN Award 2333790, NIH Bridge2AI Center Program under award 1U54HG012510-01, Cisco-UCSD Sponsored Research Project, as well as generous gifts from Google, Adobe, and Teradata. Any opinions, findings, and conclusions or recommendations expressed herein are those of the authors and should not be interpreted as necessarily representing the views, either expressed or implied, of the U.S. Government. The U.S. Government is authorized to reproduce and distribute reprints for government purposes not withstanding any copyright annotation hereon.

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

## A Appendix

### A.1 Examples of Explicitly vs. Implicitly Expressed Relationships

*Implicit relationships* do not contain class-indicative words; they are inferred through specific lexical patterns within a sentence. *Explicit relationships* are relationships that are explicitly expressed and contain class-indicative words. Figure 4 provides examples of each expression type.

| Sentence ( head , tail ) | Relation | Expression Type |
|---|---|---|
| Noam Chomsky gained the title Professor... | *title* | explicit |
| Professor Noam Chomsky of MIT's linguistics program... | *title* | implicit |
| William attended Auburn University. | *attended* | explicit |
| William (MS, Auburn University) joined the team. | *attended* | implicit |
| Philadelphia is located in Pennsylvania. | *located in* | explicit |
| The event took place in Philadelphia, Pennsylvania. | *located in* | implicit |

Figure 4: Implicitly expressed relationships are conveyed through lexical patterns, where specific linguistic patterns indicate a relationship between entities. Explicitly expressed relationships are represented through class-indicative words.

### A.2 Additional Related Work

**Continual Relation Extraction**: Continual relation extraction (CRE) is a relatively new task that focuses on continuously extracting relations, including novel relations, as new data arrives. CRE's main challenge is preventing the catastrophic forgetting of known classes (Hu et al., 2022; Zhao et al., 2022). In CRE, new data can contain known and novel classes, similar to our setting; however, CRE assumes that all new data is labeled, which fundamentally differs from our unlabeled setting.

**Zero-shot Relation Extraction:** Zero-shot relation extraction methods typically assume that test data only contains novel classes and that descriptions for those novel classes are readily available (Levy et al., 2017; Obamuyide and Vlachos, 2018; Lockard et al., 2020; Chen and Li, 2021). Generalized zero-shot relation extraction (ZSRE) removes the assumption that test data can only contain novel classes. However, ZSRE methods still heavily rely on descriptions of novel relation classes (Huang et al., 2018; Rahman et al., 2017), which is information that we do not assume is available in our unlabeled setting.

**Prompt-based RE Methods**: Prompt-based methods have shown promising results for both closed-world and open-world RE tasks (Jun et al., 2022; Wang et al., 2022a; Li et al., 2022b). Prompt-based methods for relation extraction involve constructing prompts, sometimes called "templates," that provide contextual cues for identifying relations between entities in text. These prompts typically comprise natural language phrases that capture the semantic relationship between entities. Concurrent OpenRE works Li et al. (2022b) and Wang et al. (2022a) introduce a prompt-based framework for unlabeled clustering. Prompt-based methods are used to generate relationship representations which are then clustered in a high-dimensional space. The clusters are iteratively refined using the training signal from labeled data, with careful measures to ensure the model is not biased to known classes. However, the aforementioned methods assume that unlabeled data is already divided into sets of known and novel classes, which is an unrealistic assumption of real-world unlabeled data. Furthermore, these works only report performance on novel classes, obscuring the model's overall performance in a real-world setting where the unlabeled data contains known and novel classes.

### A.3 Pre-processing and Augmenting FewRel

Special treatment is needed for FewRel dataset since it is a uniform dataset without entity type information or annotated negative instances.

**Frequency-based class splits**: To conduct the frequency-based splits described in Section 5, we obtain the distribution of relation classes as they appear in real-world data. Given that the relationship IDs in FewRel correspond with relationships in Wikipedia, we obtain class frequency information directly from Wikipedia by aggregating counts of occurrences of each relationship.

**Augmenting with negative instances**: Unfortunately, FewRel does not provide annotated negative instances (e.g., the *no relation* class). To better simulate real-world data, we augment the FewRel dataset with negative instances from ReTACRED. We recognize that augmenting FewRel with data from another dataset is not ideal since distribution differences may exist. Future work in the Generalized Relation Discovery setting may focus on extending FewRel with domain-aligned human annotated negative instances.

**Resolving entity type information**: The role of entity type information in relation extraction has been widely acknowledged (Peng et al., 2020; Wang

et al., 2022b). However, the FewRel dataset lacks explicit entity type information. To address this limitation and resolve entity types for all entities in the FewRel dataset, we employ the following two-phase approach:

1. Wikidata ontology traversal: FewRel provides a Wikidata entity ID for each entity. Leveraging the Wikidata API, we retrieve the metadata associated with each entity ID. Then, we recursively map entity types (e.g., the value of the property "subclass of" for each concept in Wikidata) to parent types until a root node is found. During this traversal, we encounter a few special cases: (1) concepts with missing values for the "subclass of" property; (2) concepts with multiple values for the "subclass of" property; and (3) values of "subclass of" that lead to looping paths in the ontology. For entities with missing values, typically found in the leaf nodes of the Wikidata knowledge graph, we default to the value of the Wikidata property "instance of" as the starting concept for our recursive transversal. When a concept has multiple values for "subclass of," we select the first value unless that value leads to a looping path within the ontology (e.g., "make-up artist" is a subclass of "hair and make-up artist," which is a subclass of a "make-up artist"). In these cases, we choose the next value of the "subclass of" until we find a non-looping path to a root node.

2. Type binning: Using the raw values of subclasses results in thousands of fine-grained entity types. We iteratively bin entity types into parent entity types until each entity type has at least 1,000 entities to obtain broader, more generalized entity types. This method produced 23 distinct entity types (see Figure 5 for the names and distribution of entity types found in FewRel).

## A.4 Baselines

In this section, we provide additional details about our baseline models.

### A.4.1 Soliciting Predictions from GPT 3.5:

GPT 3.5 often performs better on tasks with the help of in-context learning (Wei et al., 2023; Wang et al., 2023). We construct a prompt that lists all known relation classes and offers a couple examples of extracted relationships. We use natural language class names to help the model understand

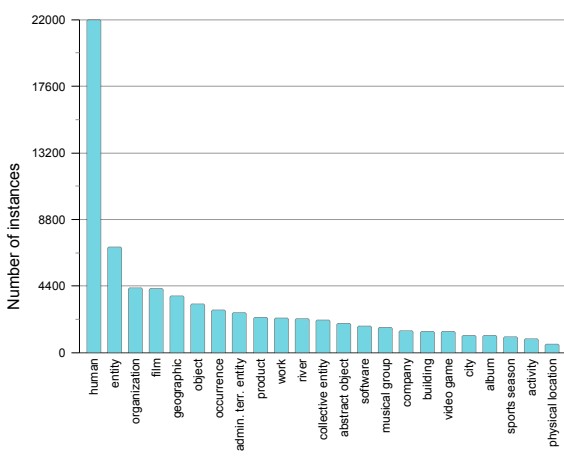

Figure 5: The distribution of the 23 entity types resulting from our recursive resolution of FewRel entity types using the Wikidata ontology.

and make predictions. The following is the prompt we used for soliciting predictions for our tests:

1. *Select the correct relation between the head and tail entities in the following unlabeled examples.*
2. *Each example has the head and tail entities appended to the sentence in the form: (head entity) (tail entity).*
3. *There are 40 known relation classes, and up to 80 unknown, or novel, relation classes.*
4. *The following is the list of known relation classes: "instance of", "subject", "language", "country", "located in", "occupation", "constellation", "citizenship", "part of", "taxon rank", "location", "heritage", "has part", "sport", "genre", "child", "country of origin", "position", "follows", "followed by", "contains", "father", "jurisdiction", "field of work", "participant", "spouse", "mother", "participant", "operator", "performer", "member of party", "publisher", "owned by", "member org", "religion", "headquarters", "sibling", "position played", "work location", "original language"*
5. *If the instance is a novel class, suggest the most likely novel class name.*
6. *Here are some examples:*
   - *In 1966 the USSR accomplished the first soft landings and took the first pictures from the lunar surface during the Luna 9 and Luna 13 missions . (Luna 13) (USSR) => ? "operator"*
   - *Her attempts to publish the work were unsuccessful until she acquired the patronage*

*of Sophia Mathilde, wife of King William III of the Netherlands. (Sophia Mathilde) (King William III of the Netherlands) => ?*
*"spouse"*

7. *Respond only with the class name in quotes: KGOR is licensed to Omaha, Nebraska United States, and serves the Omaha metropolitan area. (KGOR) (Omaha, Nebraska) => ?*

An identical prompt was also used for TACRED and ReTACRED, with changes only made to the numbers of known and novel classes, the list of known classes, and the examples provided.

### A.4.2 Probing GPT 3.5 for Prior Knowledge

One issue in evaluating GPT 3.5 is that the exact body of data used for training is unknown. Therefore, to ensure a fair comparison, we seek to determine if GPT 3.5 prior knowledge of the various datasets we use in our experiments. To do this, we ask GPT 3.5 to list all the classes in a specific dataset with the following prompt: "What are the relation classes found in the [DATASET_NAME] relation extraction dataset?" We report the response when asking about TACRED in Table 4. Note: GPT 3.5 also responded with accurate descriptions of the relation classes, but they are omitted for brevity.

For the TACRED dataset, GPT 3.5 responded with 37 correct responses of the 41 total relation classes. We use these results to argue that GPT 3.5 has an unfair advantage in discovering novel classes in TACRED and ReTACRED. Despite this advantage, GPT 3.5 did poorly compared to the other baselines we tested. However, when asked about the relation classes in FewRel, it responded with only four correct responses of the 80 total relation classes in the dataset. This information can partially explain why, in our tests, GPT 3.5 performs better on the TACRED and ReTACRED datasets compared to FewRel.

### A.4.3 A Deeper Examination of GPT 3.5's Performance

The performance of GPT 3.5 has yielded results below our initial expectations. We carefully constructed our prompts in accordance with best practices drawn from recent studies that have showcased the efficacy of in-context learning with generative models (Wei et al., 2023; Wang et al., 2023). Nevertheless, it is plausible that more effective prompting methods for open relation extraction exist. In particular, we propose exploring alternative

| Ground truth class | Response from GPT 3.5 | Type |
|---|---|---|
| org:alternate_names | org:alternate_names | TP |
| org:city_of_headquarters | org:city_of_headquarters | TP |
| org:country_of_headquarters | org:country_of_headquarters | TP |
| org:dissolved | org:dissolved | TP |
| org:founded | org:founded | TP |
| org:founded_by | org:founded_by | TP |
| org:member_of | org:member_of | TP |
| org:members | org:members | TP |
| org:number_of_employees/members | org:number_of_employees/members | TP |
| org:parents | org:parents | TP |
| org:political/religious_affiliation | org:political/religious_affiliation | TP |
| org:shareholders | org:stateorprovince_of_headquarters | TP |
| org:stateorprovince_of_headquarters | | FN |
| org:subsidiaries | org:subsidiaries | TP |
| org:top_members/employees | org:top_members/employees | TP |
| org:website | org:website | TP |
| per:age | per:age | TP |
| per:alternate_names | | FN |
| per:cause_of_death | per:cause_of_death | TP |
| per:charges | per:charges | TP |
| per:children | per:children | TP |
| per:cities_of_residence | per:cities_of_residence | TP |
| per:city_of_birth | per:city_of_birth | TP |
| per:city_of_death | per:city_of_death | TP |
| per:countries_of_residence | per:countries_of_residence | TP |
| per:country_of_birth | per:country_of_birth | TP |
| per:country_of_death | per:country_of_death | TP |
| per:date_of_birth | per:date_of_birth | TP |
| per:date_of_death | per:date_of_death | TP |
| per:employee_of | per:employee_of | TP |
| per:origin | | FN |
| per:other_family | | FN |
| per:parents | per:parents | TP |
| per:religion | per:religion | TP |
| per:schools_attended | per:schools_attended | TP |
| per:siblings | per:siblings | TP |
| per:spouse | per:spouse | TP |
| per:stateorprovince_of_birth | per:stateorprovince_of_birth | TP |
| per:stateorprovince_of_death | per:stateorprovince_of_death | TP |
| per:stateorprovinces_of_residence | per:stateorprovinces_of_residence | TP |
| per:title | per:title | TP |
| | per:countries_of_citizenship | FP |
| | per:locations_of_residence | FP |
| | per:locations_of_residence | FP |

Table 4: Comparing responses from GPT 3.5 to the ground-truth classes in the TACRED dataset. GPT 3.5. correctly predicts 37 of the 41 relation classes in TACRED, receiving an F1 score of 0.91 for class name prediction.

prompting techniques, such as Chain-of-Thought (CoT) or Self-Consistency Prompting, in future works.

To gain a more comprehensive understanding of the reasons behind GPT 3.5's suboptimal performance, we conducted an informal error analysis. Our investigation involved randomly selecting 40 instances of erroneous predictions across both known and novel classes generated by GPT 3.5. We present our observations below:

**Errors within Known Classes**: GPT 3.5's inaccuracies within known classes appear to stem from the difficulty in distinguishing between classes with subtle differences. For instance, in the context of ReTACRED, the model frequently confuses the following known classes: "org:top_members/employees,"

"org:members," and "org:member_of." Similarly, the model exhibits confusion between the "org:country_of_branch" and "org:stateorprovince_of_branch" classes. We speculate that GPT 3.5 may require a larger volume of in-context examples to discern the nuances that set these classes apart.

**Errors within Novel Classes**: Errors in predicting novel classes exhibit a common pattern in which GPT 3.5 tends to predict the nearest known and, typically, more general class, rather than suggesting a novel class name. For instance, instances of the novel class "org:shareholders" are frequently predicted as the broader but related known class "org:member_of." Furthermore, the model struggles to propose novel class names that align with novel classes, especially among novel classes that share a high degree of similarity. For instance, classes such as "per:cause_of_death," "per:city_of_death," "per:stateorprovince_of_death," and "per:country_of_death" pose challenges for the model.

The task of consistently suggesting names of novel relation classes without access to a predefined set of class names is intrinsically challenging, even for human annotators. Ideally, having access to the relationship representations produced by GPT 3.5 would allow us to leverage the advanced clustering techniques used in this paper that have proven effective in predicting novel classes. Unfortunately, the unavailability of these representations constrains our ability to employ such methods to enhance GPT 3.5's capability to predict novel classes.

#### A.4.4 Confidence-based Baselines

For our confidence-based extensions of existing OpenRE methods, we pre-train each model on the set of known classes. Then, we use a holdout set of known class instances and collect confidence scores ($c$) for each instance using the softmax function:

$$c = \frac{\exp(\mathcal{F}(\mathbf{x}, y))}{\sum_{y' \in R} \exp\left(\mathcal{F}\left(\mathbf{x}, y'\right)\right)}$$

where $\mathbf{x}$ is an input instance with relation label $y$, and $\mathcal{F}(\mathbf{x}, y))$ is a relation classifier function.

The mean of confidence scores from known instances is then used as a threshold to segment instances from unlabeled data into sets of known and novel classes—confidence scores from predictions

|  | | ReTACRED | TACRED | FewRel |
|---|---|---|---|---|
| w/o neg | RoCORE++ | 0.722 | 0.698 | 0.528 |
| | MatchPrompt++ | 0.794 | 0.712 | 0.817 |
| | TABs++ | 0.693 | 0.704 | 0.816 |
| w/neg | RoCORE++ | 0.571 | 0.551 | 0.518 |
| | MatchPrompt++ | 0.690 | 0.576 | 0.860 |
| | TABs++ | 0.474 | 0.522 | 0.737 |

Table 5: Accuracy when using the mean confidence score of labeled data to determine which instances are and are not novel from unlabeled data.

on unlabeled data that fall below the threshold are assigned novel classes and vice-versa. We report the accuracy of each pre-trained model in determining whether an instance is known or novel in Table 5. Overall, this method performs reasonably well on data without negative instances. However, with negative instances, segmenting known and novel classes becomes difficult.

#### A.4.5 ORCA Baseline

ORCA leverages a pre-trained vision model, namely ResNet (He et al., 2015), to generate representations for images. To adapt the ORCA model to the text domain, we replace ResNet with DeBERTa (He et al., 2021) and generate representations for relationships using the method used in KNoRD and described in Section 4.4. The remaining architecture and loss functions are unmodified.

### A.5 High-quality Weak Label Analysis

We retrospectively assess the effect of using different amounts ($P\%$) of high-quality weak labels generated from the GMM to train the cross-entropy module. Quality is assessed using the probability that an instance belongs to a given cluster within the GMM module. Instances within each cluster are sorted by quality, and then the top $P\%$ is selected as weak labels to train the cross-entropy model. In Table 6, We observe that, in most cases, performance is increased by selecting a subset of weak labels based on quality; however, the optimal value for $P$ fluctuates between settings. We leave determining the best $P$ value for future work set $P = 15$ for all and the experiments presented in this paper.

### A.6 Implementation Details

#### A.6.1 Prompt Model Training

In our prompt model training, we adopt the masked language modeling (MLM) task where a

| | % of Weak Labels | ReTACRED | | | TACRED | | | FewRel | | |
|---|---|---|---|---|---|---|---|---|---|---|
| | | F1 (all) | F1 (known) | F1 (novel) | F1 (all) | F1 (known) | F1 (novel) | F1 (all) | F1 (known) | F1 (novel) |
| w/o negatives | Top 10% | 0.770 | 0.927 | 0.582 | 0.667 | 0.771 | 0.582 | 0.483 | 0.514 | 0.478 |
| | Top 15% | 0.793 | 0.927 | 0.632 | 0.718 | 0.860 | 0.603 | 0.606 | 0.662 | 0.597 |
| | Top 25% | 0.758 | 0.924 | 0.560 | 0.712 | 0.806 | 0.635 | 0.537 | 0.608 | 0.526 |
| | Top 50% | 0.769 | 0.937 | 0.567 | 0.718 | 0.814 | 0.640 | 0.575 | 0.676 | 0.559 |
| | Top 75% | 0.763 | 0.933 | 0.558 | 0.721 | 0.822 | 0.640 | 0.534 | 0.707 | 0.507 |
| | All (100%) | 0.764 | 0.932 | 0.562 | 0.732 | 0.828 | 0.654 | 0.528 | 0.749 | 0.494 |
| w/negatives | Top 10% | 0.658 | 0.835 | 0.370 | 0.535 | 0.695 | 0.332 | 0.598 | 0.693 | 0.583 |
| | Top 15% | 0.677 | 0.832 | 0.434 | 0.536 | 0.685 | 0.355 | 0.583 | 0.728 | 0.559 |
| | Top 25% | 0.680 | 0.814 | 0.478 | 0.540 | 0.683 | 0.364 | 0.574 | 0.689 | 0.555 |
| | Top 50% | 0.686 | 0.824 | 0.482 | 0.553 | 0.676 | 0.405 | 0.615 | 0.723 | 0.598 |
| | Top 75% | 0.664 | 0.811 | 0.456 | 0.563 | 0.679 | 0.429 | 0.601 | 0.734 | 0.580 |
| | All (100%) | 0.673 | 0.785 | 0.499 | 0.580 | 0.672 | 0.473 | 0.584 | 0.724 | 0.561 |

Table 6: F1-micro scores reported using varied levels of high-quality weak labels with and without negative instances. Quality is measured using the probabilities assigned by the GMM that each instance belongs to a specific cluster.

`RoBERTa`[8] is trained on an NVIDIA Quadro RTX 8000 GPU.

We split labeled data into training and validation datasets by their relationships. Specifically, we hold out instances with five random relationships (excluding negative instances) from labeled data for each dataset as the validation dataset. Training with dataset splitting by relation types instead of instances can stop early before the model overfit on the known classes. The metric for validation is perplexity. Other hyper-parameters are as follows:

- learning rate: 5e-5
- batch size: 32
- probability of masking: 15%

### A.6.2 Relation Classification

All our models were trained on an NVIDIA GeForce RTX 3090 GPU. We use {41, 42, 43, 44, 45} for our seed values. We limit the length of the input sequence to 100 tokens, use a hidden dimension of 768, and use an AdamW optimizer (Loshchilov and Hutter, 2017). We use DeBERTa (He et al., 2021)[9] as the pre-trained model for all of our experiments.

We perform light hyperparameter tuning using the Optuna framework (Akiba et al., 2019). We randomly sample 20% of the instances of known classes in our labeled datasets for validation and conduct 80 trials of a hyperparameter search within the following search space:

- learning rate: [1e-4, 1e-5]
- dropout: [0.2, 0.4], (step size = 0.05)

- batch size: [64, 128], (step size = 64)
- max gradient norm: [0.8, 1.0], (step size = 0.1)
- epochs: [4, 12], (step size = 1),

Below are the final hyper-parameter settings used for each dataset:
**ReTACRED & TACRED**: batch size: 128, epochs: 5, learning rate: 1e-5, dropout: 0.2, max gradient norm: 1
**FewRel**: batch size: 128, epochs: 5, learning rate: 1e-5, dropout: 0.2

### A.6.3 Fully-supervised Training

We construct our fully-supervised data splits by assuming all classes are known and using the method described in Section 5. We combine instances from labeled and unlabeled data and randomly selected 15% to form the test split. The remaining instances were used for training, with 20% of the training data further segmented as the validation set.

Note that our results from the fully supervised setting cannot be directly compared to numbers reported on popular benchmarking websites[10] since our splits do not match the standard. Our splits are designed to maintain consistency with our other experiments within the proposed Generalize Relation Discovery setting.

---

[8]https://huggingface.co/roberta-base
[9]https://huggingface.co/microsoft/deberta-base

[10]https://paperswithcode.com/sota