# OpenReview forum: "Open-world Semi-supervised Generalized Relation Discovery Aligned in a Real-world Setting"
_EMNLP/2023/Conference — EMNLP 2023 Main_

### Official Review · Reviewer_ZBDY · 2023-07-30

**Soundness:** 4

**Excitement:**

4: Strong: This paper deepens the understanding of some phenomenon or lowers the barriers to an existing research direction.

**Missing References:**

N/A

**Paper Topic And Main Contributions:**

This paper studies a more general setting in OpenRE, Real-world OpenRE. This is an interesting and important question worthy of study. The authors observe that popular relations can often be implicitly inferred through specific patterns, while long-tail relations tend to be explicitly expressed. Based on this observation, this paper proposes the KNoRD and conducts extensive experiments to evaluate its effectiveness.

**Questions For The Authors:**

1. Please add a detailed explanation and comparison of the work of ARD [1] in your final version. Even if the setting you are concerned about are different from it, I think it is necessary to use a certain amount of space to explain the difference between the settings you are concerned about and the setting that ARD is concerned about. Kindly, if you can take this issue seriously enough in your rebuttal and final version, I'm willing to change my rating.
2. Why is the performance of ChatGPT very poor? This result seems counter-intuitive, I guess it has something to do with the prompt you designed. I think this also needs to be analyzed and explained.

[1]. Active Relation Discovery: Towards General and Label-aware Open Relation Extraction

**Reasons To Accept:**

1. The issues and settings that this paper focuses on are important and meaningful.
2. This paper is well-written.
3. The experiments are sufficient and combined with large models such as ChatGPT for analysis.

**Reasons To Reject:**

1. My main concern with this paper is the Generalized Relation Discovery setting it claims to propose, which is actually not the first time the authors have proposed it, and has been proposed in earlier work such as ARD. Unfortunately, I can not see enough comparison and elaboration of ARD in this paper.

**Reproducibility:**

5: Could easily reproduce the results.

**Reviewer Confidence:**

4: Quite sure. I tried to check the important points carefully. It's unlikely, though conceivable, that I missed something that should affect my ratings.

**Typos Grammar Style And Presentation Improvements:**

There are no punctuation marks after each equations, so you need to pay attention.

---

> ### Author Rebuttal · Authors · 2023-08-24
>
> **The ARD method**: Thank you for the thoughtful comment about the ARD model. Although we mention the ARD work in our Related Work section (lines 193-197) and briefly discuss the differences of the setting used for ARD compared to the setting we propose, we do not provide specifics on why we do not compare our method to the ARD method. Below, we provide some additional details and, if accepted, we will add a summarized version of this explanation to our manuscript.
>
> The ARD method is a three-part method that (1) develops relation representations, (2) conducts relational outlier detection to detect known and novel classes, and, finally, (3) conducts relational active learning where human annotators annotate instances of novel classes which are then used to train a classifier. Part 3 of the ARD model is fundamentally different from our semi-supervised setting. All of the baselines we compare against, as well as the method we propose, are semi-supervised and do not require human supervision beyond the instances provided in the labeled dataset. By providing additional human annotations of novel instances, we argue that the ARD approach aligns better with continual learning and few-shot learning approaches. The adaptation needed to allow the ARD method to function within our setting requires a significant number of design decisions that we feel we are unable to make. Indeed, such a modified method would likely require a considerable amount of exploration.
>
> **ChatGPT performance**: We agree, the performance from GPT 3.5 is significantly lower than expected. It is possible that a better prompt method exists to increase the performance. We do our best to design our prompt using the best practices from some recent studies that show the effectiveness of in-context learning with generative models [1,2].
>
> We speculate that the low performance from GPT 3.5 results from the difficulty in determining the granularity of the relation classes of interest. Many of the responses generated from GPT 3.5 were not technically wrong, they were just of a different level of granularity. For example, GPT 3.5 often predicted the relation class “member of” instead of the more specific and correct class “shareholders.” For novel classes, however, we believe the task of predicting the name of a relation class without having a set of class names to choose from is very challenging even for humans. Ideally, we would have access to the relationship representations produced by GPT 3.5, then we would be able to apply advanced clustering techniques that have proved effective in predicting novel classes. Unfortunately, these representations are not available so we are limited in the methods we can use to help GPT 3.5 predict the correct novel classes.
>
> We believe this area of research is very interesting and deserves a deeper analysis. Alternative prompting methods such as chain-of-thought  (CoT) or self-consistency prompting should be explored in future works. We will include an error analysis of the prompt method we use with GPT 3.5 in the Appendix of our revised draft to hopefully inspire deeper insights and analysis into LLMs applied to the OpenRE task.
>
>
> [1] Larger language models do in-context learning differently (2023)
>
> [2] Large Language Models Are Implicitly Topic Models: Explaining  and Finding Good Demonstrations for In-Context Learning (2023)

---

### Official Review · Reviewer_sj25 · 2023-08-04

**Soundness:** 4

**Excitement:**

4: Strong: This paper deepens the understanding of some phenomenon or lowers the barriers to an existing research direction.

**Paper Topic And Main Contributions:**

The paper works on open-world relation extraction and mainly consider the set of novel classes including long-tailed relations.

The authors believe long-tail relations tend to be explicitly expressed, and based on this motivation, they propose method composed of prompt-based training, semantically aligned representation, Clustering with Majority-vote Bifurcation and relation classification, using DeBERTa.

The proposed method is compared with several state-of-the-art open-world relation extraction baselines. The proposed method outperforms these baselines with big margins.

**Questions For The Authors:**

1. Long-tailed relations still have some samples that can be used. In problem statement, why there is totally no samples for novel classes to predict? Could you justify how the samples of the long-tail relations are used? Or are their samples just given up?

**Reasons To Accept:**

1. The paper is well written. The technical quality is good, with a method well designed and well presented.

2. The results are quite promising, with outperformance over the baselines, on three datasets (TA-CREAD, ReTACRED, and FewRel).

**Reasons To Reject:**

1. The assumption of no distribution shift is a bit strong. It's likely that known relation classes found in labeled data do not appear in the unlabeled data.

2. Could you justify the construction of novel classes in 385 - 389? E.g., why the ratio 15% is used? Is there any previous standard setting on the evaluation of this problem?

**Reproducibility:**

3: Could reproduce the results with some difficulty. The settings of parameters are underspecified or subjectively determined; the training/evaluation data are not widely available.

**Reviewer Confidence:**

3: Pretty sure, but there's a chance I missed something. Although I have a good feel for this area in general, I did not carefully check the paper's details, e.g., the math, experimental design, or novelty.

**Typos Grammar Style And Presentation Improvements:**

In 304, "the same feature-space" to "the same feature space".

In 256, "Semantically-aligned" to "Semantically aligned".

---

> ### Author Rebuttal · Authors · 2023-08-24
>
> We would like to thank you for your thoughtful feedback and comments. We will fix the minor typos you mentioned in our revised draft and, below, we address your major comments. If accepted, we will include some of the added details below in our updated manuscript.
>
> **No distribution shift**: For the set of known classes, we select the top 50% most frequently appearing classes in the labeled data. While the absence of known classes within unlabeled data is a plausible scenario, we believe it is likely that known classes would manifest in unlabeled data owing to their prevalence. By following a previous work in computer vision (ORCA [1]) and assuming no distribution shift between labeled and unlabeled data, we confine our inquiry to a singular domain of data. Allowing for a distributional shift between labeled and unlabeled data fundamentally transforms the nature of the problem into one of detecting out-of-domain relationships which, interestingly, has been criticized as a limitation of previous OpenRE works since it is unlikely that new, unlabeled data exclusively contains novel classes.
>
> Furthermore, by assuming no distribution shift between labeled and unlabeled data, we construct what we believe is a more difficult and more realistic setting. A significant challenge in our setting is the differentiation between known and novel classes. We report performance across both categories of class types—known and novel—asserting the broader applicability of this approach to real-world scenarios.
> 15% ratio: The 85/15 train/test split of instances belonging to known classes was inspired by the previous work (TABs [2]). We believe other ratios, specifically a lower ratio of training data (e.g. 50/50) would increase the problem’s difficulty and would be very interesting to explore in future works.
>
> **Samples for long-tail relations**: If we understand the question correctly, samples from novel, long-tail classes are not used to train the prompt model or the cross-entropy model directly because we have no labels for the novel instances. We do, however, generate pseudo-labels for the novel classes via clustering and “majority vote bifurcation” modules and, subsequently, use those pseudo-labels to train the cross-entropy model, along with gold labels from the labeled dataset.
>
>
> [1] Open-world semi-supervised learning (2022)
>
> [2] Open Relation and Event Type Discovery with Type Abstraction (2022)

---

### Official Review · Reviewer_h8YH · 2023-08-05

**Typos Grammar Style And Presentation Improvements:** In line 546, figure 3 caption, "uncon…
**Soundness:** 4

**Excitement:**

4: Strong: This paper deepens the understanding of some phenomenon or lowers the barriers to an existing research direction.

**Paper Topic And Main Contributions:**

The paper proposes the KNoRD method for discovering implicit and explicit known and novel relations from unlabeled data. In contrast to the general OpenRE setup, authors make no simplifying assumptions on the type of relation classes (known, novel, and negative) in the chosen dataset. The KNoRD method is evaluated on three well-known relation extraction datasets, along with a detailed ablation analysis on each involved component.


**Questions For The Authors:**

Question A: In line 223, each sentence x_j in the unlabeled data contains an entity pair. Does this mean all the sentences in the unlabeled data have an entity pair? If this is true, then to extend the KNoRD method to new unlabeled data, should one already perform entity recognition and filter out the valid sentences?

Question B: As mentioned in footnotes 2 and 3, there is a difference between the number of relation [MASK] tokens used during training and inference time. How does the model perform well during the inference time? Or do the majority of the relation tokens map to a single [MASK] token during training, and the rest have no effect?

Question C: How are the top three tokens from the vocabulary chosen for unconstrained relation representation (line 313)?

Question D: What was the intuition behind choosing 85% and 15% split for known vs novel relations? Won't it make more sense to follow the actual frequency distribution of the relation classes and decide on the threshold for the split? E.g., The train relation frequency distribution of the three chosen datasets could be helpful.

**Reasons To Accept:**

The paper is well-written and easy to understand. The assumptions and claims for building the KNoRD model are clearly stated. It's laudable that the motivation behind creating the KNoRD model is to handle the realistic real-world data setting. The KNoRD model has the potential to generate weak data labels, and it would be interesting to use it for synthetic data creation.

**Reasons To Reject:**

The KNoRD method works only on sentence inputs, can't easily be extended to document-level RE, and requires human-annotated labeled data. There are various hyperparameters (top-3 tokens for relation representation, P% for weak labels in clustering, known cluster size, etc.) used throughout the KNoRD modeling, which could make it hard to reproduce the results or even apply to a new unlabeled dataset.

**Reproducibility:**

4: Could mostly reproduce the results, but there may be some variation because of sample variance or minor variations in their interpretation of the protocol or method.

**Reviewer Confidence:**

4: Quite sure. I tried to check the important points carefully. It's unlikely, though conceivable, that I missed something that should affect my ratings.

---

> ### Author Rebuttal · Authors · 2023-08-24
>
> Thank you for your questions and comments. Below are responses to each of your questions. If accepted, we will summarize and add these responses to the manuscript to increase the paper’s clarity.
>
> **Question A**:  Your statement is correct -- all sentences are assumed to have two or more entities that are either pre-identified by the dataset or by an entity resolution module. To extend KNoRD to new, unlabeled data, one would need to first resolve entities and only consider sentences with two or more found entities. In constructing our problem setting, we follow previous works (MatchPrompt [1], RoCORE [2], TABs [3]) by specifically focusing on relation extraction separate from entity recognition; we assume access to resolved entities in both labeled and unlabeled data. We argue that this makes the results more modular in the sense that the reported performance of the relation extraction and relation discovery is not dependent on the performance of the entity resolution module. We could have removed this assumption and approached the problem through a multi-task learning lens but, given the difficulty of our current setting, we believe this modification would be best suited for future works when more progress has been made on the current setting.
>
> **Question B**: For training, the number of tokens that the model predicts corresponds to the number of tokens found in the ground truth relationship name. However, as you mention, during inference time, the model only predicts a single token. This is to account for the fact that, without labels, we do not have access to the ground truth relation name for each instance. We use the top three predicted singular tokens for each instance and average them together to both mitigate the effects of selecting a single token and to create a richer, more robust relationship representation during our clustering phase.
>
> **Question C**: The top three tokens are selected by simply taking the top three most likely tokens as determined by the output probabilities from the prompt model for each inputted instance.
>
> **Question D**: To clarify, the 85/15 split refers to the split within the set of known classes – 85% of instances from known classes are used for training, and 15% of known class instances are used for testing. The ratios for this split follow a previous OpenRE work (TABs [3]). We use a 50/50 split between known and novel classes where the top 50% most frequent classes are assigned to the set of known classes and the remaining classes are assigned to the set of novel classes. Given that our novel classes are exclusively long-tail classes with few instances, we selected a large 50/50 known/novel class split in hopes of creating a representative test set that contained a relatively large amount of novel instances. However, this choice was one we made, and, as you correctly identify, there are other methods we could have used. We could have used the actual frequency distribution of the relation classes in each dataset to determine the threshold. However, it is our understanding that instances of relationships in real-world data very often form a long-tail distribution. We did not include graphs that show this but the three datasets we used all have similar-looking long-tail distributions. We believe that using the class frequency of each dataset to determine the threshold of the splits would have resulted in similar splits.  We can provide these graphs in our updated draft to further justify our decision to do a 50/50 known and novel class split.
>
>
> [1] MatchPrompt: Prompt-based Open Relation Extraction with Semantic Consistency Guided Clustering (2022)
>
> [2] A Relation-Oriented Clustering Method for Open Relation Extraction (2021)
>
> [3] Open Relation and Event Type Discovery with Type Abstraction (2022)

---

### Meta-Review · Area_Chair_LZHd · 2023-09-21

**Recommendation:** 5

**Metareview:**

This paper focuses on open-world relation extraction problem, and emphasizes novel classes including long-tailed relations. Experiments are generally solid, with large model-based analysis. All 3 reviewers reached a consensus on strong soundness and excitement.

---

### Decision · Program_Chairs · 2023-10-07

**Decision:**

Accept-Main

**Comment:**

This paper focuses on open-world relation extraction problem, and emphasizes novel classes including long-tailed relations. Experiments are generally solid, with large model-based analysis. All 3 reviewers reached a consensus on strong soundness and excitement.